# Ephs and Ephrins in Adult Endothelial Biology

**DOI:** 10.3390/ijms21165623

**Published:** 2020-08-06

**Authors:** Dianne Vreeken, Huayu Zhang, Anton Jan van Zonneveld, Janine M. van Gils

**Affiliations:** Einthoven Laboratory for Vascular and Regenerative Medicine, Department of Internal Medicine, Leiden University Medical Center, 2333 ZA Leiden, The Netherlands; h.zhang@lumc.nl (H.Z.); a.j.van_zonneveld@lumc.nl (A.J.v.Z.); j.m.van_gils@lumc.nl (J.M.v.G.)

**Keywords:** Eph–Ephrin signaling, endothelial cells, blood vessels

## Abstract

Eph receptors and their ephrin ligands are important guidance molecules during neurological and vascular development. In recent years, it has become clear that the Eph protein family remains functional in adult physiology. A subset of Ephs and ephrins is highly expressed by endothelial cells. As endothelial cells form the first barrier between the blood and surrounding tissues, maintenance of a healthy endothelium is crucial for tissue homeostasis. This review gives an overview of the current insights of the role of ephrin ligands and receptors in endothelial function and leukocyte recruitment in the (patho)physiology of adult vascular biology.

## 1. Introduction

The Eph super family is the largest family of receptor tyrosine kinases (RTKs) in humans. Since their initial discovery in 1987 [1], erythropoietin-producing hepatocellular receptors (Ephs) and their Eph receptor interacting protein (ephrin) ligands have been shown to be involved in a plethora of physiological and pathological processes. While originally discovered during embryonic development in the patterning of the nervous system, the Eph family is also crucial for embryonic vasculo- and angiogenesis [2]. The best known and first to be uncovered ephrin family members are ephrinB2 and one of its receptors—Eph receptor B4 (EphB4). EphrinB2 and EphB4 are differentially expressed in arterial and venous endothelial cells, respectively, and are essential for cardiovascular development [3,4]. In recent years, it has become increasingly clear that the functions of ephrins go beyond embryogenesis. Besides remaining functional in the adult brain where they are involved in the remodeling of neuronal circuits, Eph family members can modulate angiogenesis, immunoregulation, bone maintenance and glucose and intestinal homeostasis [2,5]. Resulting from having such a broad range of regulatory functions, ephrins are also associated with several pathologies e.g., cancer and inflammatory diseases such as fibrosis and atherosclerosis [5]. However, the precise contribution of Ephs and ephrins in established vessels and the endothelium in adults remains incompletely understood.

As the first interface between circulating blood and surrounding tissues, endothelial cells are the main regulators of vascular homeostasis. A healthy endothelium comprises a single-cell layer of endothelial cells covered with a glycocalyx. Endothelial cells form a proper barrier with a low and selective permeability to fluids, solutes and blood cells by means of a dynamic interplay between its cellular cytoskeleton, interendothelial cell–cell junctions and cell–matrix interactions. Maintaining a healthy endothelium is an important process involving regulated tissue regeneration and remodeling, regulation of endothelial permeability, (inflammatory) cell trafficking, vascular tone and blood coagulation. A dysfunctional endothelium, characterized by a condition associated with impaired bioavailability of nitric oxide, a proinflammatory and procoagulant phenotype, can result in impaired endothelial function including increased permeability to both fluid and cells and loss of vascular tone. A dysfunctional endothelium can elicit a plethora of diseases such as e.g., cancer metastasis, diabetes and cardiovascular disease [6]. This review will focus on the role of endothelial ephrin ligands and receptors, and in particular on their effect on endothelial function and their role in leukocyte recruitment, in the physiology and pathology of adult vascular biology.

## 2. Ephrins and Ephs Basic Structure and Signaling

### 2.1. Ephrins and Ephs Basic Structure

The Eph family is the largest family of RTKs comprising 14 Eph receptors and 8 ephrin ligands in mammals, which are both membrane bound. Ephrins and their receptors are subdivided into classes A and B based on membrane attachment and sequence homology and ligand preference, respectively. The A-class ephrin ligands (ephrinA1-ephrinA5) are characterized by a glycosylphosphatidylinositol (GPI) anchor to the cell membrane, while B-class ephrins (ephrinB1-ephrinB3) have a transcellular and cytoplasmic domain with a PSD-95/Dlg/ZO-1 (PDZ)-binding motif (Figure 1A). The Eph receptors are transmembrane proteins with an extracellular domain that contains a ligand binding domain, a cysteine rich region and two fibronectin type II domains. Its intracellular domain contains a juxtamembrane region containing two tyrosine residues, a protein tyrosine kinase domain, a sterile alpha motif (SAM) domain and a PDZ-binding domain. While the original concept was that EphA receptors bind mainly to ephrinA ligands and EphB receptors to ephrinB ligands, more recent research has shown that receptor–ligand interactions can also occur between opposite classes [7,8].

### 2.2. Forward Signaling

In contrast to all other RTKs, ephrins and their receptors can induce bidirectional signaling. Binding of the ligand to the receptor induces forward signaling of the receptor, while reverse signaling of the ligand is initiated upon the binding of the receptor to the ligand. By initiating signaling, ephrins can induce multiple signaling cascades and influence several cellular processes. Upon ligand binding, Eph receptors cluster and are activated by autophosphorylation of tyrosine residues within the juxtamembrane region and the kinase domain of the receptor. This phosphorylation enables the recruitment of a variety of Src-homology 2 (SH2) domain-containing adaptor proteins that can induce further downstream signaling pathways including mitogen-activated protein kinases (MAPK), small GTPases, focal adhesion kinases (FAK) and protein kinase B (AKT). The association of PDZ-domain proteins with Eph receptors also contributes to the signaling of activated Eph receptors. Most of the cellular pathways activated by ephrin forward signaling are involved in the regulation of the cytoskeletal dynamics of cells and therewith can modulate cellular processes such as migration, adhesion and proliferation [7,8]. Examples of signaling events that can occur upon Eph/ephrin interaction and clustering are shown in Figure 1B.

### 2.3. Reverse Signaling

The initiation of ephrin reverse signaling is primarily described for ephrinB ligands. As mentioned before, these ephrinB ligands have a transmembrane domain that contains several phosphorylation sites and a PDZ-binding domain. Upon ligand–receptor interaction, the intracellular domain of ephrinB ligands is phosphorylated by Src Family Kinases (SFKs) enabling the binding of adaptor proteins, such as Grb4. In addition, PDZ-domain proteins, such as PDZ-RGS3, can bind to activated ephrinB ligands and induce, for example, an interaction with G-protein-coupled receptors or the activation of small GTPases. Both these phosphorylation-dependent and phosphorylation-independent ways of ephrinB signaling result in the regulation of the cellular cytoskeleton [7,9].

The downstream reverse signaling mechanisms of ephrinA ligands are less clear. EphrinA ligands lack a transmembrane domain and are only anchored to the cell membrane with a GPI anchor and therefore lack clear mechanisms to induce intracellular signaling. However, there is some evidence that ephrinA ligands make use of transmembrane binding partners, such as integrins, the RTK tropomyosin receptor kinase B (TrkB) and rearranged during transfection (Ret) to convey their extracellular signal and modulate e.g., cellular adhesion and apoptotic cell death (Figure 1B) [8,10].

### 2.4. Alternative Signaling

Besides the best characterized way of Eph/ephrin signaling, bidirectional signaling, ephrins and Eph receptors can also induce signaling in other ways. Ephrins and their receptors can (1) signal independently from each other, (2) undergo cis interactions between receptors and ligands on the same cell and (3) they can recruit other signaling partners to enforce or prevent their signaling. Ephrin signaling is complex and can activate many different pathways, sometimes even with contrasting effects. The expression of Eph receptors and their ligands on a cell itself (cis interactions) and its neighboring cells (trans interactions), the amount of Eph/Ephrin clustering and the variety of receptors and ligands involved in these clusters are only some of the factors that determine Eph/Ephrin downstream signaling events. The eventual outcome of Eph/ephrin signaling will be the sum of all cellular and microenvironmental components in a certain cell at a certain time [8,11]. More detailed information on Eph/ephrin signaling can be found in previous reviews [7,8,11].

## 3. Ephrin and Eph Expression in Endothelial Cells

The currently available literature on the expression and regulation of expression of ephrins and the Eph receptors in endothelial cells are described in this section and summarized in Table 1 and Table 2.

### 3.1. Ephrins and Ephs Expressed under Homeostasis

The expression of ephrins and its Eph receptors in vivo in healthy human endothelium has mainly been investigated for ephrinB2 and EphB4. As mentioned before, ephrinB2 and EphB4 are differentially expressed in arteries and veins, with ephrinB2 marking arteries and EphB4 marking veins [3,4], a property that is not only important during embryonic development but persists through adulthood [32,33]. Despite the ”clear” distribution of EphB4 and ephrinB2 expression in veins and arteries in vivo, this distinction is no longer observed in vitro [20]. Both arterial and venous endothelial cells have been shown to express ephrinB2 and EphB4. At the protein level, a lot less information has been reported. However, for EphB2, EphB4 and ephrinB2, protein has been detected in human umbilical vein endothelial cells (HUVECs), human aortic endothelial cells (HAECs) and human dermal microvascular endothelial cells (HDMECs) [34]. EphrinB1 protein was observed only in human coronary artery endothelial cells (HCAECs) [35].

Providing a nice overview of ephrin RNA expression in endothelial cells, Sakamoto et al. showed that all Eph and ephrin genes except EphA1 and EphB3 are expressed by HCAECs. High levels of expression have been observed for ephrinA1, ephrinA4, ephrinA5, ephrinB1, ephrinB2, EphA2, EphA4, EphB1, EphB2 and EphB4 [35]. Although most of the highly expressed genes are confirmed in other studies and/or in different endothelial cell types such as HUVECs, HAECs, HDMECs and human lung microvascular endothelial cells (HLMVECs), there are some observations that are less unambiguous [12,34,36,37]. For example, the expression of ephrinB1. While the expression of ephrinB1 is detectable in HCAECs, it is not observed in HAECs and HDMECs [34] and only sometimes in HUVEC [34,36]. In addition, a considerable amount of variation is observed in the lesser expressed genes. For example, EphB3 has been shown to be expressed in HUVEC, HAEC and HDMEC, while it could not be detected in HCAECs [35]. In contrast, EphA3-A6, EphA8, ephrinA2 and ephrinA3 could be detected in HCAECs [35] but not in HAECs and HUVEC [12]. Therefore, despite a considerable amount of overlap in ephrin (receptor) expression between different endothelial cells, some variation exists. The organ of origin of the endothelial cells, the method of culturing and the method of detection are probably the main contributors to these varying observations.

### 3.2. Ephrin and Eph Regulation by Inflammation and Hemodynamic Factors

The expression of ephrin ligands and their receptors can be modulated by several environmental factors. Inflammation has been shown to upregulate the expression of the Eph family members ephrinB1 and ephrinB2 [19,21], while EphB4 expression is not regulated by the inflammatory cytokines interleukin (IL)-1β nor tumor necrosis factor (TNF)-α [31]. On the other hand, ephrinA1 and EphA2 expression is increased upon stimulation with inflammatory cytokines [12,19]. Vascular endothelial growth factor (VEGF), which is upregulated by several inflammatory cytokines, can also increase the expression of EphA2 [38] and ephrinB2 [20]. Inflammation does not only regulate the expression of several ephrins and their receptors, ephrins can also regulate the expression of inflammation-related genes. For example, the addition of ephrinA1 to HAECs or HUVECs significantly upregulated the expression of vascular cell adhesion molecule (VCAM)-1 and E-selectin or chemokine (C-C motif) ligand (CCL) 2 and chemokine (C-X-C motif) ligand (CXCL) 1, respectively [12,39].

Next to inflammation, hemodynamics is an important regulator of ephrin and Eph expression. Most hemodynamic ephrin research has been performed investigating ephrinB2 and its receptor EphB4. Results regarding the effect of shear on ephrinB2 are not uniform. While a few studies reported a decrease in ephrinB2 in endothelial cells exposed to high laminar shear stress [21,26], others reported no effect [22] or even an upregulation of ephrinB2 expression [23]. Differences in flow rates and the use of more premature endothelial cells might be causative for the observed differences. In vivo experiments in mice showed that the disruption of normal perfusion, by the ligation of an artery, induced expression of ephrinB2 [22]. An ex vivo experiment with human saphenous vein segments showed a trend of upregulation of ephrinB2 when exposed to arterial shear stress [24]. Similar to under inflammatory conditions, EphB4 expression seems to be less dependent on shear stress. Besides a slight upregulation of EphB4 after 4 h of high arterial shear stress, over time, EphB4 levels remained stable [26]. In contrast, in more premature cells, EphB4 expression was decreased upon arterial shear stress [23]. Additionally, ex vivo arterial shear stress decreased the expression of EphB4 in the human saphenous vein, while venous shear stress did not alter EphB4 expression [24]. One study showed a downregulation of ephrinA1 upon laminar shear stress. Exposure to turbulent flow even further decreases the expression of ephrinA1 compared to laminar shear stress [15].

### 3.3. Ephrin and Eph Regulation by Other Environmental Conditions

As indicated earlier, the asymmetrical arteriovenous expression of ephrinB2 is not maintained in vitro, indicating that for retaining the differential arteriovenous expression profile of ephrinB2 in endothelial cells, microenvironmental cues are necessary. Indeed, changing the microenvironment of cultured HUVECs by co-culturing them with smooth muscle cells has resulted in an upregulation of ephrinB2 [20]. In addition, Eph/ephrin expression depends on cell culture conditions. For example, the expression of ephrinA1 is highly dependent on cell density and serum as high cell density and serum depletion both induce an increase in the expression of ephrinA1 [13]. In addition, altering oxygen levels can induce changes in ephrin expression. Hypoxia induced expression of ephrinA3 (via microRNA-210 downregulation, see more below) [16] and oxygen–glucose deprivation/reperfusion, as a simulation of ischemic conditions, induced the expression of both ephrinA1 and EphA4 [14].

### 3.4. Ephrin and Eph Regulation by MicroRNA’s

With the increasing awareness regarding the role of microRNA’s (miRNAs) in the cellular response to environmental signals, the regulation of endothelial ephrins by miRNAs has also been investigated. miRNAs are small non-coding RNAs that, by binding to complementary sequences, mostly in the 3′ UTRs of target mRNAs, can regulate the expression of these genes. In most cases, miRNAs suppress target mRNA expression, but opposite forms of regulation have been described as well [40]. While most miRNAs localize intracellularly, they can also be released from cells via extracellular vesicles and, upon fusion of the vesicles with target endothelial cells, regulate endothelial gene expression in a paracrine, endocrine and autocrine manner [41,42]. By targeting a multitude of endothelial cell genes such as CXCL12, nitric oxide synthase 3 (NOS3) and VCAM-1, miRNAs have been shown to affect endothelial cell (dys)function related processes such as barrier function, vascular tone and leukocyte trafficking [41,43].

The regulation of ephrins by miRNAs has been described for the ephrinA3 ligand, where increased levels of miR-210 repressed the protein expression of ephrinA3, and decreased levels of miR-210 increased ephrinA3 expression [16,17,18]. Expression of the EphA2 receptor was shown to be regulated by miR-26a, where a mimic of miR-26a decreased EphA2 expression, and the use of a miR-26a inhibitor increased EphA2 expression [28]. In addition, ephrinB2 and EphB4 are described to be potentially regulated by miR-20b [27], EphB2 and EphB4 by miR-520h [30] and EphA7 is described as a direct target of miR-137 [29].

## 4. Ephrins and Ephs in Endothelial Cell Proliferation and Apoptosis

### 4.1. Ephrins and Ephs in Endothelial Cell Proliferation

To maintain a heathy endothelium, the vascular system relies both on the regulated replacement of dysfunctional endothelial cells via apoptosis and cell proliferation of the neighboring cells as well as on re-endothelialization by bone marrow-derived endothelial progenitor cells (EPCs) [44]. Several ephrin family members have been described to regulate cellular proliferation and/or apoptosis. EphrinA1 reverse signaling inhibits endothelial cell proliferation, as the overexpression of ephrinA1 decreased proliferation while a knockdown increased the proliferation rate of endothelial cells [13]. The effect of EphA2 forward signaling on proliferation rates in endothelial cells is more controversial. Stimulation of the EphA2 receptor with recombinant ephrinA1 showed an upregulation of pro-survival and proliferation associated genes such as VEGF receptor 2 (VEGFR-2) and VEGFR-3 in HAECs [12]. However, the prevention of EphA2 forward signaling via a knockdown of EphA2 also induced endothelial cell proliferation [13], while primary microvascular endothelial cells of EphA2 knockout mice in culture did not show any differences at all in proliferation rates nor in apoptosis [45].

Results for the effect of ephrinB2-induced receptor signaling on proliferation are also quite contradictory. Some studies showed an (dose-dependent) increase in proliferation of endothelial cells when exposed to ephrinB2 [46,47], while others showed no effect [48] or even a decrease in proliferation of endothelial cells when grown on a surface of immobilized ephrinB2 [49]. Loss of the EphB4 receptor resulted in decreased proliferation [50]. The use of different proliferation models, different cell types and, probably most importantly, different dosages and forms of ephrinB2 ligand presentation (apical vs. basolateral, clustered vs. unclustered), could be causative for these contradicting observations. While the precise effect and mechanisms of ephrinB2 on proliferation are unclear, there are indications that the effect is mediated via the PI3K/Akt/NO/PKG/MAPK pathway [46]. In addition, while these papers suggest the involvement of the EphB4 receptor to relay the ephrinB2 signal, none of them could definitively exclude the role of other Eph receptors.

Studies regarding Eph/ephrin signaling in re-endothelialization by means of EPCs are limited. However, the downregulation of EphA2 via miR-26a could impair EPC function [51], while ephrinB2 has been shown to both enhance [52,53] and to reduce EPC function [54,55].

### 4.2. Ephrins and Ephs in Endothelial Cell Apoptosis

In addition to the proliferative functions described above, ephrinA3, ephrinB1/B2/B3 and EphB3 have been shown to play a role in apoptosis. First, decreased expression of ephrinA3, regulated by miR-210, abolished angiotensin II induced apoptosis [18]. H_2_O_2_-induced cell death of endothelial cells could be inhibited by exposing cells to the ephrinB2 ligand as indicated by increased survival rates, an increased expression of the (anti-apoptotic) B-cell lymphoma 2 (Bcl-2) gene and a decrease in caspase-3 cleavage [47]. Exposure of HUVECs to ephrinB3 also, at least partly, prevented apoptosis induced by growth factor removal. In addition, the administration of ephrinB3 in a mouse injury model resulted in less apoptosis of cortical vascular endothelial cells and a less dramatic decrease in vessel density after injury. However, knockout of the EphB3 receptor rendered similar results and, therefore, the EphB3 receptor was described to function as a pro-apoptotic dependence receptor, where ligand binding or receptor knockout is necessary to prevent EphB3 from inducing cellular apoptosis [56]. On the other hand, the knockdown of the ephrinB1 and ephrinB3 ligands in endothelial cells induced apoptosis and ephrinB1 even selectively more in senescent cells compared to nonsenescent cells [57].

## 5. Ephrins and Ephs in Endothelial Cell Adhesion, Spreading and Migration

### 5.1. EphrinA Family Members in Endothelial Cell Migration

Ephrins also regulate processes such as endothelial cell adhesion, cell spreading and migration. Most of the current available data are on the ephrinB ligands and EphB receptors. However, a few papers also indicate a role for ephrinA ligands and EphA receptors in endothelial cell adhesion and migration. When ephrinA1 is coated on a surface, it acts as a repulsive cue and prevents cell migration to the areas coated with ephrinA1 [13]. Reduced expression of ephrinA1 or EphA2 on endothelial cells themselves promoted cell migration by increasing migration velocity via the modulation of focal adhesions. Overexpression of ephrinA1 also resulted in the increased migration but via promoting the cells’ directionality instead of its velocity [13]. In contrast, a study of Rhodes et al. showed that a reduction in ephrinA1 in pulmonary endothelial cells inhibited adhesion and migration as well as the formation of capillary-like structures [58]. However, the activation of EphA2 with soluble ephrinA1 promoted Transwell migration as well as vascular assembly of endothelial cells and is mediated by RAC-1/PI3K signaling [45].

EphrinA3 has also been shown to modulate endothelial cell migration. Under normoxic conditions, the expression of ephrinA3 prevents endothelial migration, while the downregulation of ephrinA3 via the hypoxia-induced miR-210 increased endothelial transmembrane migration [16]. Cells with a decreased expression of ephrinA3 also form significantly less capillary-like structures, while the overexpression of ephrinA3 enhances tube formation [17].

### 5.2. EphrinB Family Members on Endothelial Cell Adhesion and Spreading

Similar to ephrinA1, surfaces containing ephrinB1 or ephrinB2 ligands show a repulsive effect on the adhesion of endothelial cells. EphrinB1 on its own does not have much effect but in combination with a fibronectin and nitrocellulose pre-coating [59] or when multimerized before stimulation [60], ephrinB1 acts as a repulsive cue for endothelial adhesion. An ephrinB2 coating dose-dependently inhibits endothelial adhesion of both mouse [49] and human endothelial cells, and the inhibition can even be enhanced with pre-clustering of the ephrinB2 ligands [61]. Endothelial detachment was also observed in 3D endothelial cell/smooth muscle cell co-culture spheroids and umbilical vein explants stimulated with soluble ephrinB2 [61]. In addition, ephrinB2 coating suppressed endothelial cell spreading [49], and the stimulation of endothelial cells with ephrinB2 induced cellular retraction via a mechanism dependent on both Cdc42 and Rho GTPase signaling [62]. Surface coating with the Eph receptors EphB1 or EphB4 seems to have the opposite effect, as the Eph receptor coating had no effect or dose dependently promoted endothelial cell adhesion, depending on the origin of the endothelial cells [49,61,63].

### 5.3. EphrinB Forward Signaling on Endothelial Migration

The effect of ephrinB ligands and receptors on endothelial migration is more complex. With the ability of bidirectional signaling of the ephrin family, the variety of available assays to study migration and the range of variability within these assays makes it challenging to combine all existing data. The presence of the soluble ephrinB2 ligand has been shown to significantly diminish the distance covered by endothelial cells after 48 h [61]. If present in the upper well of the Transwell system, ephrinB2 can inhibit endothelial cell migration after 4 h, and this inhibition is even more pronounced when chemotaxis was induced by the presence of VEGF in the lower well [61]. Additionally, the presence of ephrinB ligands in the lower well of the Transwell system inhibits endothelial migration, but only in combination with VEGF [48] or cultured stem cells from the apical papilla [64]. EphrinB2-induced forward signaling was also shown to inhibit endothelial sprouting and network formation [61], while the reduction in EphB4 signaling increased the formation of capillary-like structures [50].

In sharp contrast, other studies show an increase in endothelial cell migration in the presence of ephrinB2 after 24 h [47] or when stimulated with preclustered ephrinB1 for 6 h [65]. Pre-stimulation of HUVECs for 10 min with either ephrinB1 or ephrinB2 also enhanced CXCL12-induced endothelial cell transmigration after 16 h [34], and the exposure of endothelial cells to ephrinB1 multimers promoted tube formation [60]. In addition, abolishing EphB receptor forward signaling by reducing receptor expression with either antisense oligonucleotides, siRNAs, specific EphB2/EphB4 inhibiting peptides or lithocholic acid significantly diminished endothelial migration and the formation of capillary-like structures [34,66,67,68]. When the ephrinB ligand is present in the lower well of a Transwell system, a small majority of papers show an (dose-dependent) increase in endothelial cell migration across matrix-coated filters, which seems to be dependent on PI3K/AKT/MMP signaling [46,47,69]. The reduction in EphB4 signaling by a partial knockdown of EphB4 in mouse lung endothelial cells decreased in vitro migration towards ephrinB2 as well as towards serum [50], also suggesting a promigratory role for EphrinB2/EphB4 signaling in endothelial cell migration. Differences in culture conditions such as the presence or absence of serum, the method of inducing migration (scratching vs. the use of inserts vs. Transwell migration) and the duration of the experiments are most likely responsible for the differences observed.

### 5.4. EphrinB Reverse Signaling on Endothelial Migration

While the effect of Ephrin/EphB forward signaling on migration is ambiguous, reverse signaling induced by soluble EphB proteins is primarily promigratory. Exposure of endothelial cells to soluble EphB2 [63] or EphB4 [61] promoted endothelial lateral migration, and the overexpression of ephrinB2 resulted in further and faster migration of endothelial cells, as measured by time-lapse microscopy [70]. In addition, the disturbance of ephrinB reverse signaling by mutating its signaling domains, either its phosphotyrosine-dependent or PDZ-dependent signaling, results in a, respectively, mild to severe inhibition of endothelial migration further supporting a promigratory effect of ephrinB reverse signaling in endothelial cells [71]. However, the availability of EphB4 in the upper well of a Transwell system [61] or a coating of EphB4 [49] do not affect migration rates. In contrast, EphB4-induced reverse signaling does promote endothelial sprouting and formation of capillary-like structures [61], while the abrogation of reverse signaling by a knockdown of ephrinB2 in endothelial/mesenchymal stem cell co-culture models resulted in less capillary-like structures [64,72]. As the Eph family clearly affects endothelial adhesion, spreading and migration, a role for Ephs and ephrins in blood vessel remodeling is undeniable. The role of ephrins in arterio- and angiogenesis has already been discussed at length previously, and detailed overviews can be found in the following reviews: [9,73].

## 6. Ephrins and Ephs in Endothelial Barrier Function

### 6.1. EphA2 Forward Signaling Induces Vascular Leakage

Besides processes as cellular adhesion and migration, Ephs and ephrins have also been shown to influence the integrity of the endothelium by e.g., regulation of endothelial barrier permeability. VEGF has long been known to be an important regulator of the endothelial barrier, but how it regulates remains poorly understood [74]. In a study of Miao et al., it was shown that VEGF activates the intracellular PI3K/Akt and Erk1/2 signaling pathways resulting in the increased expression of EphA2 which then, in turn, contributes to an increase in paracellular permeability [38]. The role of (ephrinA1-induced) EphA2 forward signaling in reducing the endothelial barrier function has also been shown in several other in vitro and in vivo studies, where the increased expression of EphA2 increases vascular permeability, and reduced expression decreases (ephrinA1-induced) vascular permeability [28,37,39,75]. Opposite to the increase in vascular permeability upon EphA2 forward signaling, the exposure of endothelial cells to EphA4 recombinant protein could protect the endothelial barrier against TNFα-induced vascular leakage. However, whether this effect is due to blocking ephrin ligand/receptor interactions or by inducing ephrin reverse signaling is not clear [76].

### 6.2. EphrinB/EphB Signaling in Vascular Leakage

Activation of endothelial ephrinB2 reverse signaling with soluble EphB2 or EphB2 overexpressing mouse myeloma cells showed a translocation of vascular endothelial cadherin (VE-cadherin) and enhanced endothelial permeability [19]. While stimulation with EphB4 also induces a translocation of ephrinB2 itself, it does not affect the adherens junction protein VE-cadherin, implying no change in vascular integrity and permeability [20,77].

EphB4 receptor signaling has been shown to be important in preventing endothelial leakage, as low levels of EphB4a associated with edema formation and disorganization of zonula occludens-1 (ZO-1) positive endothelial junctions in a zebra fish model [78]. In addition, a recent paper of Luxan et al. showed that EphB4 is important for the structural integrity of endothelial cells, as the knockdown of EphB4 resulted in impaired cell–cell junctions and decreased cell stiffness in HUVECs in vitro and capillary ruptures in vivo. Interestingly, this paper also shows the involvement of EphB4 in transport functions of endothelial cells as reduced levels of EphB4 disrupted caveolar function and lipid transport [79]. Knockout of the EphB1 receptor also decreased caveolae formation as EphB1 no longer protected against the degradation of caveola-associated proteins [80].

## 7. Ephrins and Ephs in Leukocyte-Endothelial Cell Interactions

Besides endothelial cells, Eph receptors and ephrins are also expressed on several leukocytes. For example, primary monocytes and the human monocytic cell line THP-1 cells have been shown to strongly express ephrinA3, ephrinA4, EphB2, EphB3, EphB4 and EphB6 [35,77,81]. A T-lymphocyte cell line was shown to strongly express, among others, ephrinA1, ephrinA3, ephrinA4, ephrinB1, ephrinB2, EphA4, EphB2, EphB3, EphB4 and EphB6 [35,77]. Expression of the Eph family on both endothelial cells and leukocytes implicates a role for them in leukocyte–endothelial cell interactions as Eph–Ephrin signaling can induce changes in the leukocytes as well as in the endothelial cells.

### 7.1. EphrinA/EphA-Mediated Leukocyte Adhesion and Migration

The activation of ephrinA ligand signaling in T-lymphocytes induces the integrin-mediated adhesion of T-cells to integrin ligands as well as endothelial cells, while the activation of EphA receptor signaling reduced the adhesion of lymphocytes [82]. Additionally, T-lymphocyte migration is affected by, at least, ephrinA1. However, the direction of regulation is less clear since it has been shown to both inhibit [83] as well as increase [84] T-lymphocyte chemotaxis towards a CXCL12 gradient via cytoskeletal rearrangements. This difference could most likely be explained by the fact that one study uses immobilized ephrinA1 while the other used soluble dimeric protein.

When looking at the endothelial side, increased ephrinA1 availability, either by the addition of recombinant ephrinA1 protein or endothelial overexpression, makes the endothelium more prone to adhesion of leukocytes, while a reduction in endothelial ephrinA1 results in decreased adhesion [12,85,86]. Both EphA2 and EphA4 forward signaling can increase monocyte adhesion via modulating the surface expression of intercellular adhesion molecules (ICAM)-1 and VCAM-1 [12,85,87], induced via calcium/calcineurin dependent activation of the transcription factor nuclear factor of activated T-cells (NFAT) 1 [88]. The transactivation of EphA2 via thrombin-activated protease-activated receptor (PAR)-1 also increases ICAM-1 expression and therewith promotes leukocyte adhesion [89]. In addition, activation of the EphA4 receptor by ephrinA1 can also induce cytoskeletal rearrangements via induction of the Rho signaling pathway [86].

### 7.2. EphrinB/Eph-Mediated Leukocyte Adhesion and Migration

Not only the ephrinA subclass of the ephrin family but also ephrinB family members have been shown to modulate leukocyte–endothelial interactions. For example, a surface coating of ephrinB2 enhances monocyte adhesion [90], while both soluble ephrinB2 and ephrinB1 had no effect on monocyte adhesion [81,90]. Lymphocyte migration, on the other hand, is diminished when membranes are coated with ephrinB2/B1 [83,91], while preincubation with ephrinB1 or ephrinB2 used as a chemoattractant increased migration [21,92]. Pre-incubation with antibodies against ephrinB1/B2 inducing reverse signaling, increased T-cell migration, while a double knockout impaired their migration [93]. Independent of its ephrinB1/B2 ligands, reduction in EphB2 expression in a monocyte cell line induces cytoskeletal changes resulting in decreased monocyte adhesion and migration [81].

In combination with endothelial cells or ex vivo aortic segments with a high or low expression of ephrinB2, adhesion of monocytes is increased or decreased, respectively [19,77,90]. In addition, endothelial overexpression of ephrinB2 also enhances transendothelial migration of monocytes, which could be even further enhanced when monocytes also overexpressed the EphB4 receptor [77]. Exposure of endothelial cells to recombinant ephrinB/EphB protein also promotes lymphocyte adhesion and transmigration. Like ephrinA1, ephrinB2 can activate the EphA4 receptor, inducing Rho GTPase mediated cytoskeletal rearrangements that promote monocyte adhesion [94]. Activation of ephrinB2 by either EphB2 or EphB4 guides leukocytes to or modulates the integrity of endothelial junctions, facilitating leukocyte transendothelial migration [19,77]. Activation of endothelial ephrinB1 with EphB2 enhances leukocyte migration via a JNK-dependent upregulation of the adhesion molecules E-selectin and VCAM-1 [19].

## 8. Ephrins and Ephs and Disease

Due to their different functions in not only endothelial cells but also several other cell types, ephrin family members have been implicated in several diseases. Of these diseases, cancer is probably the best studied. Because of their roles in angiogenesis, proliferation, cell survival and cell motility, ephrins and Ephs are involved in several stages of tumor progression such as tumor angiogenesis and metastasis. In addition, ephrins are also involved in a variety of other diseases including neurological disorders and viral infections [2,95].

A healthy endothelium is crucial for the proper functioning of the vascular system and thereby also for organ perfusion. Therefore, it is not surprising that a dysfunctional endothelium underlies many (chronic) diseases such as cardiovascular disease, ischemia reperfusion injury and associated organ failure. As described above, ephrins are involved in several facets of endothelial cell function and therefore also modulate endothelial cell-related diseases.

### 8.1. Vascular Leakage and Ischemia Reperfusion Damage

In line with ephrins regulating vascular permeability, several diseases linked to vascular permeability show involvement of ephrin family members. For example, the increased vascular permeability observed in different forms of induced lung injury in mice could be reduced by an EphA2 knockout, administration of recombinant EphA2 or EphA2 receptor blocking antibodies [39,96,97]. The breakdown of the blood-brain-barrier after traumatic brain injury or induced by cerebral malaria in mice, also marked by increased vascular leakage, could be prevented by knockout of EphB3 [56] or EphA2, respectively [75], while the induction of Eph signaling by administration of ephrinA1 after ischemia/reperfusion injury induced blood-brain-barrier damage. These ephrinA1-treated mice showed more inflammation and edema in the brain and had decreased neurological scores. The ephrinA1-induced effects could be prevented when ephrinA1 and recombinant EphA4 were administered together [14]. Induction of ephrinB2/EphB4 signaling prevents against vascular leakage and neurological damage after mild cerebral ischemia by increasing endothelial-pericyte interactions [98]. Administration of recombinant EphA4-Fc also decreased vascular leakage and neutrophil accumulation after intestinal ischemia/reperfusion [76]. Besides EphA4, EphA2 has also been implicated as a potential strategy to protect against mesenteric ischemia reperfusion injury, as high dosages of EphA2-Fc or a small molecule receptor antagonist, both preventing Eph/Ephrin forward signaling, also protected against vascular permeability and inflammation after intestinal ischemia reperfusion. While recombinant Eph receptors are used here as an inhibitor of Eph/ephrin interactions by preventing ligand-induced receptor forward signaling, the potential effect of recombinant receptor administration on ephrin reverse signaling is not discussed [99].

### 8.2. Atherosclerosis

Following the fact that ephrins are involved in many processes related to atherosclerosis such as leukocyte adhesion and migration and vessel permeability, it is not surprising that several articles imply a role for ephrins in atherosclerosis. Not only is the expression of ephrins regulated under pro-atherosclerotic conditions such as turbulent flow and exposure to pro-inflammatory cytokines, several studies have also shown expression of ephrins within plaque lesions. For example, ephrinA1, ephrinB1, EphA2, EphA4 and EphB2 have been found in human atherosclerotic plaques [12,81,86,91]. Additionally, ephrinB2 has been found in atherosclerotic lesions but its expression does not significantly differ from expression in relatively normal vessels [24]. However, in atheroprone regions such as the inner curvature of the aorta expression of ephrinB2 is elevated compared to the outer curvature [21,90]. A particularly interesting finding is the fact that the Eph genes EphA2, EphA8 and EphB2 are located within the murine Athsq1 (atherosclerosis) susceptibility locus [100], which is highly homologous to the premature myocardial infarction susceptibility locus in humans that similarly contains EphA2, EphA8 and EphB2 [101]. Additional studies have shown a pro-atherosclerotic role for EphA2 and EphB2. In vivo knockout of EphA2 in ApoE-deficient mice showed decreased atherosclerotic lesion formation and less inflammation [87,102], while decreased expression of EphB2 in monocytes results in decreased adhesion and migration of these cells in vitro [81]. While all these studies seem to indicate a potential role, additional research is necessary to show the precise mechanisms of ephrins on the development of atherosclerosis.

## 9. Conclusions

Ephrin ligands and receptors have long been known as regulators of neuronal and vascular development. This review summarizes evidence for a sustained role for the ephrin family of RTKs in adult endothelial biology. Endothelial cells are key players in vascular homeostasis and therefore crucial for sufficient delivery of nutrients and oxygen to tissues. This review shows the involvement of primarily ephrinA1 and its main receptors, EphA2 and EphA4, and ephrinB1/B2 and its receptors, EphB2 and EphB4, in vascular homeostatic functions such as endothelial cell renewal, migration, barrier integrity and leukocyte interactions via regulation of the cellular cytoskeleton and cell–cell junctions (Figure 2).

Despite the growing amount of research, the precise contributions of ephrins in adult endothelial cells remain largely inconclusive. While gene expression of a large number of ephrin ligands and Eph receptors was detected in endothelial cells, most up-to-date studies focus on the highly expressed ephrinA1/B1/B2 ligands and EphA2/A4/B2 and B4 receptors, while others are barely studied. The contribution of other (moderate) expressed ephrin ligands and receptors but also their potential interplay with other ligands and receptors will be of great interest for gaining a better understanding of ephrin ligand/receptor function in endothelial behavior.

On a more functional level, due to the fact that (1) the ephrin family signals bidirectionally, (2) they can affect a broad variety of downstream intracellular binding partners and signaling pathways, (3) their signaling is dependent on the full spectrum of ephrins available, (4) ephrin receptors and ligands are highly interchangeable and (5) they are expressed in multiple cell types, the characterization of the precise role for the individual ligands and receptors is hampered. Besides this variability in ephrin receptor/ligand interactions and signaling, differences in culture conditions and experimental setups are also major contributors to the observed variability in ephrin effects. For example, the use of different migration assays, scratching versus silicon inserts versus Transwell migration, already affects endothelial behavior. In addition, differences in serum levels, timing of experiments and method of ephrin ligand or receptor activation and/or inhibition could lead to the opposite effects on endothelial function observed for some ephrin ligand/receptor interactions.

All data combined still favor the idea that ephrin ligand and receptor signaling is important for regulating endothelial function, even though the precise direction and associated mechanisms are not always obvious (yet). Future research pursuing more uniform and physiological relevant forms of research will contribute to discriminating between the different roles and interactions of ephrin ligands and receptors and their single or shared contributions in mature endothelial homeostasis. A better understanding of the ephrin family in vascular health but also in pathophysiology requires more in-depth research and is of great interest for unraveling novel targets to prevent or alleviate endothelial dysfunction and related diseases.

## Figures and Tables

**Figure 1 ijms-21-05623-f001:**
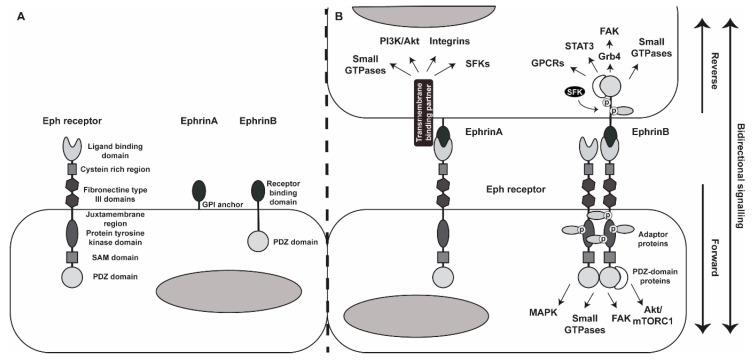
Eph/Ephrin structure and signaling (**A**) general structure of the Eph receptors, ephrinA ligands and ephrinB ligands. (**B**) Eph/Ephrin bidirectional signaling and some of its signaling pathways. SAM = sterile alpha motif, GPI = glycosylphosphatidylinositol, PI3K = phosphoinositide 3-kinase, AKT = protein kinase B, SFK = Src family of kinases, GPCR = G protein-coupled receptor, STAT3 = signal transducer and activator of transcription 3, FAK = focal adhesion kinases, MAPK = mitogen-activated protein kinase, mTORC1 = mammalian target of rapamycin complex 1, *p* = phosphor.

**Figure 2 ijms-21-05623-f002:**
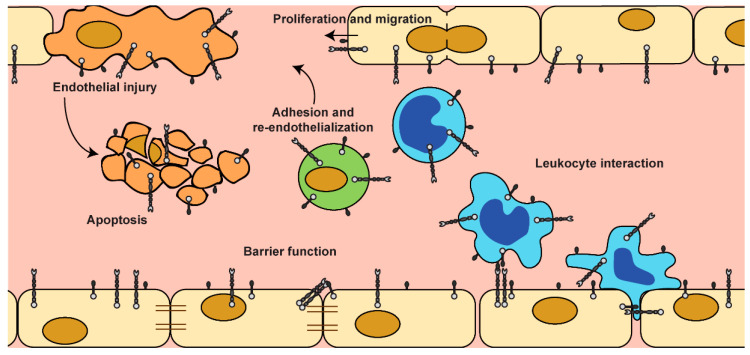
Schematic overview of Eph/Ephrin function in adult vascular biology.

**Table 1 ijms-21-05623-t001:** Ephrin ligand expression in endothelial cells.

Ephrin Ligands	Endothelial Cells						(Patho)physiological Conditions
	HCAECs	HAECs	HUVECs	HCAECs	HDMECs	HLMVECs	
EphrinA1	High	High	High	High		High	Increased by inflammation [12], increasing cell density [13], serum depletion [13], ischemia [14]. Decreased by shear stress [15].
EphrinA2	Moderate	Low/no	Low/no	Low/no			
EphrinA3	Moderate	Low/no	Low/no	Low/no			Increased by hypoxia [16]. Decreased by miR-210 [16,17,18].
EphrinA4	High	Moderate	Moderate	Moderate			
EphrinA5	High	Moderate	Moderate	Moderate			
EphrinB1	High	Undetected	Moderate		Undetected		Increased by inflammation [19].
EphrinB2	High	High	High		High		Increased by inflammation [19,20,21], laminar or interrupted flow [22,23,24,25]. Unchanged by laminar flow [22]. Decreased by laminar flow [21,26], miR-20b [27].
EphrinB3	Moderate	Undetected	Moderate/no		Undetected		

**Table 2 ijms-21-05623-t002:** Eph receptor expression in endothelial cells.

EphReceptors	Endothelial Cells	(Patho)physiological Conditions
HCAECs	HAECs	HUVECs	HCAECs	HDMECs	HLMVECs
EphA1	Undetected	Low/no	Low/no	Low/no			
EphA2	High	High	High	High		High	Increased by inflammation [12,20].Decreased by miR-26a [28].
EphA3	Moderate	Low/no	Low/no	Low/no			
EphA4	High	High	Moderate	High			Increased by ischemia [14].
EphA5	Moderate	Low/no	Low/no	Low/no		Moderate	
EphA6	Moderate	Low/no	Low/no	Low/no		Moderate	
EphA7	Moderate	Low/no	Low/no	Low/no			Decreased by miR-137 [29].
EphA8	Moderate	Low/no	Low/no	Low/no			
EphA10		Low/no	Low/no	Low/no			
EphB1	High	High	High		Moderate		
EphB2	High	High	High		Moderate		Decreased by miR-520h [30].
EphB3	Undetected	Moderate	High		Moderate		
EphB4	High	High	High		High		Decreased by inflammation [23], flow [23,24], miR-20b [27], miR-520h [30]. Unchanged by flow [31].
EphB6	Moderate	Moderate	Moderate		Moderate

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
