# Peer review of "Ephs and Ephrins in Adult Endothelial Biology"

_ijms, 2020, doi:10.3390/ijms21165623_

Round 1

Reviewer 1 Report

Eph receptors and their ephrin ligands play a crucial role in different physiological and pathological conditions. A group of Ephs and their ligands are largely expressed by endothelial cells  which represent the first barrier between blood and surrounding tissues and consequently play a crucial role in tissue homeostasis. Thus, the review focuses mainly on the role of Ephs and their ligands in endothelial functions.

The review is interesting and well organized I recommend publication after minor revisions.

Line 181 “the expression these genes” should be “the expression of these genes”

Table 1 "Increased by hypoxia[31]" add a comma; “Increased by inflammation [20]” add a full stop

Line 426 “not only is expression of ephrins” should be “not only expression of ephrins is”

Line 439 “while all indicating” I’d say “while all these studies seem to indicate”

Line 458 “their binding partners are not 1:1 correlated” this sentence is not clear- what do the authors mean with 1:1 correlated?

Author Response

July 2020

RE: Manuscript ID: ijms-885831 - Ephs and ephrins in adult endothelial biology

Reply to the reviewer:

We thank the reviewer for his/her comments. We have addressed his/her concerns and implemented these in our current manuscript. Please find our response to the reviewer below.

Minor comments:

  1. Line 181 “the expression these genes” should be “the expression of these genes”.

Thank you for pointing this out. We adjusted this is our manuscript.

  1. Table 1 "Increased by hypoxia [31]" add a comma; “Increased by inflammation [20]” add a full stop.

As requested we added the indicated comma and full stop and checked and corrected both tables for consistent use of commas and full stops.

  1. Line 426 “not only is expression of ephrins” should be “not only expression of ephrins is”.

The sentence was revised according suggestion.

  1. Line 439 “while all indicating” I’d say “while all these studies seem to indicate”.

We agree, this adjustment is a better description. We have adjusted as requested.

  1. Line 458 “their binding partners are not 1:1 correlated” this sentence is not clear- what do the authors mean with 1:1 correlated?

In response to the reviewers comment, the 1:1 correlation was supposed to describe that Ephrin ligands and receptors have a high variety of downstream binding targets that can bind and induce several downstream signaling pathways. Not only do both Eph receptors and ephrin ligands contain different binding domains, e.g. PDZ domain and phosphorylation sites, but each cell contains its own set of downstream binding partners, such as Src family kinases and mitogen-activated protein kinases, depending on e.g. cell type and environmental exposures. To improve the clarity of this point we have changed the sentence “their binding partners are not 1:1 correlated” to “they can affect a broad variety of downstream intracellular binding partners and signaling pathways” (Page 11, line 467-469).

Reviewer 2 Report

Introduction:

Please provide a proper definition of endothelial dysfunction.

Paragraph 3.4. Ephrin and Eph regulation by microRNA’s:

The Authors should better describe the mechanistic role of miRNAs in endothelial (dys)function:

Int J Mol Sci. 2020 Mar 16;21(6):2012.

J Cell Mol Med. 2019 Dec;23(12):7933-7945.

J Cell Physiol. 2016 Aug;231(8):1638-44. doi: 10.1002/jcp.25276

Int J Mol Sci. 2019 Jan 5;20(1):172. doi: 10.3390/ijms20010172

Author Response

July 2020

RE: Manuscript ID: ijms-885831 - Ephs and ephrins in adult endothelial biology

Reply to the reviewer:

We thank the reviewer for his/her comments. We have addressed his/her concerns and implemented these in our current manuscript. Please find our response to the reviewer below.

Major comments:

  1. Please provide a proper definition of endothelial dysfunction.

We are grateful for the reviewers comment and agree that the manuscript will improve with providing a proper definition of endothelial dysfunction. We consider endothelial dysfunction as a pathological state of endothelial cells resulting in impaired function. Endothelial dysfunction is characterized by a condition associated with impaired bioavailability of nitric oxide, a proinflammatory and procoagulant phenotype, loss of vascular tone and increased vascular leakage (of both fluids and cells). We have included this definition in our review (Page 1, line 41-44).

  1. Paragraph 3.4. Ephrin and Eph regulation by microRNA’s: The Authors should better describe the mechanistic role of miRNAs in endothelial (dys)function.

Thank you for pointing this out and for the provided literature suggestions. As this paragraph mainly focused on the regulation of ephrin expression, we indeed kept the introduction of miRNA’s relatively short. We now added a more elaborate description of the general function of miRNA’s on endothelial cell function (Page 5, line 183-193).

Round 2

Reviewer 2 Report

-